# Deep Reinforcement Learning Car-Following Control Based on Multivehicle Motion Prediction

**Tao Wang** [1,2], **Dayi Qu** [1,*], **Kedong Wang** [1,3] and **Shouchen Dai** [1]

1   School of Mechanical and Automotive Engineering, Qingdao University of Technology,
    Qingdao 266520, China; wangtaoqd1987@163.com (T.W.); 18661831717@163.com (K.W.);
    dsc2177@163.com (S.D.)
2   School of Artificial Intelligence and Big Data, Zibo Vocational Institute, Zibo 255300, China
3   Intelligent Manufacturing Institute, Qingdao Huanghai University, Qingdao 266427, China
*   Correspondence: dayiqu@qut.edu.cn

**Abstract:** Reinforcement learning (RL)–based car-following (CF) control strategies have attracted significant attention in academia, emerging as a prominent research topic in recent years. Most of these control strategies focus solely on the motion status of the immediately preceding vehicle. However, with the development of vehicle-to-vehicle (V2V) communication technologies, intelligent vehicles such as connected autonomous vehicles (CAVs) can gather information about surrounding vehicles. Therefore, this study proposes an RL-based CF control strategy that takes multivehicle scenarios into account. First, the trajectories of two preceding vehicles and one following vehicle relative to the subject vehicle (SV) are extracted from a highD dataset to construct the environment. Then the twin-delayed deep deterministic policy gradient (TD3) algorithm is implemented as the control strategy for the agent. Furthermore, a sequence-to-sequence (seq2seq) module is developed to predict the uncertain motion statuses of surrounding vehicles. Once integrated into the RL framework, this module enables the agent to account for dynamic changes in the traffic environment, enhancing its robustness. Finally, the performance of the CF control strategy is validated both in the highD dataset and in two traffic perturbation scenarios. In the highD dataset, the TD3-based prediction CF control strategy outperforms standard RL algorithms in terms of convergence speed and rewards. Its performance also surpasses that of human drivers in safety, efficiency, comfort, and fuel consumption. In traffic perturbation scenarios, the performance of the proposed CF control strategy is compared with the model predictive controller (MPC). The results show that the TD3-based prediction CF control strategy effectively mitigates undesired traffic waves caused by the perturbations from the head vehicle. Simultaneously, it maintains the desired traffic state and consistently ensures a stable and efficient traffic flow.

**Keywords:** car-following; reinforcement learning; twin-delayed deep deterministic policy gradients; sequence-to-sequence; motion prediction

## 1. Introduction

With the continuous advancements in advanced driver assistance systems (ADASs) such as adaptive cruise control, cooperative adaptive driving control, lane-keeping assistance, and emergency brake assistance, the advent of a transformative era in autonomous transportation is imminent [1]. As an essential function of ADASs, adaptive cruise control, which is closely related to the CF strategy, is instrumental in improving driving comfort, lessening driver strain, enhancing precision in vehicle handling, and bolstering vehicular safety. Routinely executed by drivers, the task of CF constitutes a fundamental component of driving activities and has been the subject of various CF models designed to encapsulate human driving behaviors.

Traditionally CF models have primarily focused on interactions between two vehicles: a following vehicle and a lead vehicle [2]. This two-vehicle-mode perspective has shaped

classical car-following research, providing insights into how drivers respond to the actions of vehicles directly ahead. However, real-world driving necessitates awareness beyond the immediate lead vehicle, especially when considering braking scenarios and the broader context of vehicular networks where information exchange can substantially enhance situational awareness [3].

The advent of high-resolution traffic data, coupled with the advances in artificial intelligence, has ushered in a new era of CF behavior modeling, redefining the landscape with an unprecedented level of precision and insight [4,5]. Through the amalgamation of comprehensive field data and data-driven strategies, this innovative paradigm extends beyond the confines of traditional modeling techniques, embracing a multitude of variables to forge models of unparalleled depth and relevance. Particularly noteworthy is the adoption of artificial neural networks (ANNs) [6–8] and recurrent neural networks (RNNs) [9–13], which epitomize the synthesis of advanced pattern recognition and sequential data analysis. These methodologies not only refine the granularity of driver behavior predictions but also enhance the models' adaptability, transcending the limitations of their predecessors. Additionally, incorporating RL into CF models allows them to adapt dynamically, more closely mimicking the complexity of real driving situations [14–19]. Overall, these advancements bring us closer to understanding and simulating driver behavior accurately.

The development of CF models in transportation research has achieved notable advancements, yet these models still face some limitations that restrict their effectiveness in dynamic and complex driving environments. Traditional CF models, which depend on a limited set of parameters and rely on empirical calibration, lack the necessary adaptability for varying traffic conditions, potentially leading to issues with model generalization. Likewise, while data-driven approaches excel at capturing detailed vehicle interactions, their reliance on extensive historical datasets can result in suboptimal performance, often influenced by biases or unrepresentative samples of driving behaviors.

In contrast, the integration of V2V communication introduces a transformative aspect to CF modeling, particularly beneficial in scenarios where autonomous vehicles (AVs) interact within mixed traffic flows. This enhancement allows AVs to extend their situational awareness, integrating data from both leading and following vehicles to refine decision-making processes, optimize speed control, and enhance overall traffic safety.

Amidst these advancements, RL emerges as a pioneering approach, distinguishing itself from conventional methodologies by facilitating an interactive, dynamic learning environment. This attribute is especially pivotal in crafting advanced control strategies that align with the nuanced and unpredictable nature of real-world driving.

To mitigate the identified shortcomings of existing CF models, this research introduces a predictive deep reinforcement learning (DRL)–based CF control strategy that underscores some improvements:

1.  By focusing on multivehicle scenarios, the strategy provides an exhaustive understanding of traffic dynamics, enabling more informed and efficacious decision making that boosts traffic safety and efficiency.
2.  The implementation of the twin-delayed deep deterministic policy gradient (TD3) algorithm address the issues of prediction accuracy and learning stability, fostering more reliable CF behavior modeling.
3.  The integration of a seq2seq model within the TD3 framework facilitates a more sophisticated prediction of surrounding vehicles' movements, increasing the model's adaptability and accuracy across various traffic situations.

Through these concerted efforts, this study aims to refine CF modeling within the realm of autonomous driving, advancing our understanding and application of intelligent transportation systems by leveraging the synergies of advanced AI methodologies and enhanced vehicular communication technologies.

## 2. Multivehicle CF Scenarios

Classical CF control strategies primarily focus on a two-vehicle scenario involving the SV and the immediately preceding vehicle. Indeed, vehicles trailing the SV are also crucial in real driving scenarios, particularly when the SV brakes due to emergencies. With the development of perception and communication technologies, the vehicle behind the SV can transmit detailed information, allowing for an accurate status perception. In connected environments, CAVs can receive data from several vehicles ahead and behind using V2V technology, enhancing traffic operational efficiency and reliability. Therefore, the multivehicle CF scenarios considered in this study are shown in Figure 1.

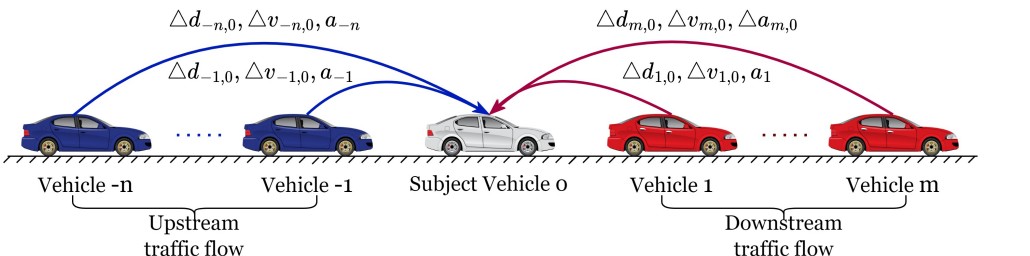

**Figure 1.** The multivehicle CF scenario.

Consider an open, flat, single-lane scenario. In this setup, the SV is indexed as 0. The preceding vehicles in the downstream traffic flow are represented by the set $\mathcal{L} = \{1, 2, ..., m\}$, and the vehicles trailing the SV in the upstream traffic flow are indexed using the set $\mathcal{F} = \{-1, -2, ..., -n\}$. In a connected environment, the SV is able to gather the motion statuses of the surrounding vehicles, which always include relative distance $\triangle d_{i,0}$, relative velocity $\triangle v_{i,0}$, and acceleration $a_i$ of each vehicle $\forall i \in \mathcal{L} \cup \mathcal{F}$. The SV then makes informed decisions based on this information.

## 3. Methodology

### 3.1. CF Control Framework

Figure 2 shows the framework for a CF control strategy, which is based on RL and the prediction of the surrounding vehicles' motion statuses. Utilizing a seq2seq network, the acceleration kinematic states of surrounding vehicles, represented as $\hat{a}_i(t + 1 : t + H)$, $\forall i \in \mathcal{L} \cup \mathcal{F}$, are predicted, thereby enabling a proactive adaptation to the dynamic traffic environment at each time step, with CF scenarios extracted from the highD dataset. In this refined RL framework, the prediction of surrounding vehicles' motion statuses employs a sophisticated network that utilizes both real-time and historical data facilitated by V2V communication technology. The subsequent control strategy rooted in this framework operates within an interactive RL environment, wherein the agent, guided by the actor-critic network and the TD3 policy algorithm, seeks to optimize its decisions to enhance the reward outcomes, hence updating the environment to the succeeding state effectively.

The integration of the predictive mechanism with an RL-based control strategy highlights the nuanced application of the highD dataset to scrutinize and interpret complex multivehicle dynamics. The methodological approach adopted herein delicately navigates beyond the conventional CF analysis by embedding a multivehicle statuses predictive model within the RL framework. This fosters the development of an agent endowed with sophisticated decision-making capabilities.

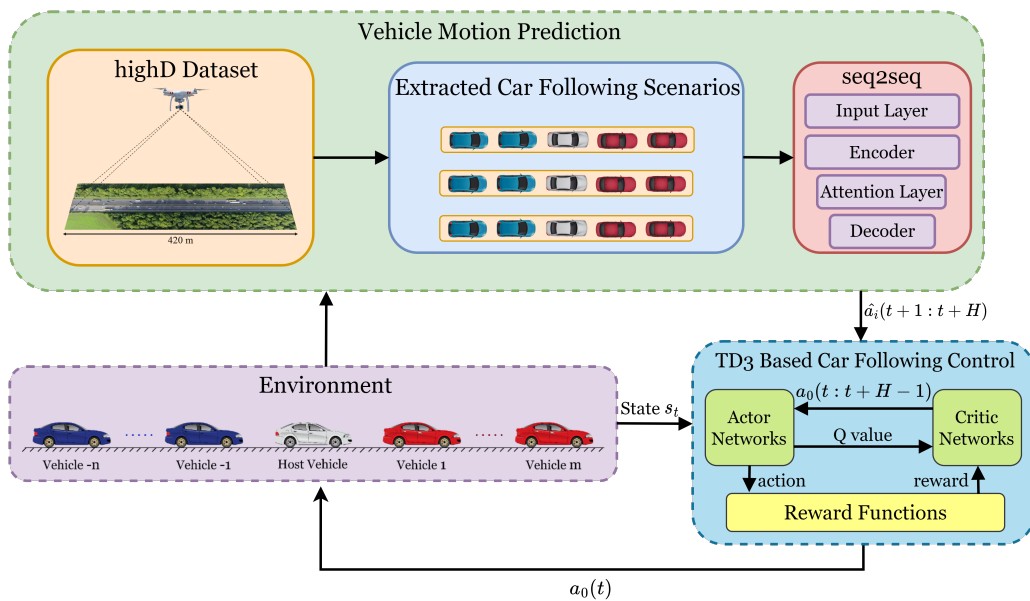

**Figure 2.** Car-following control framework.

### 3.2. Vehicle Motion Prediction Model

To comprehensively account for the diverse influences on driving behavior and develop an intelligent model for predicting future vehicle motion statuses, a refined seq2seq deep learning architecture is proposed. This model focuses on harnessing historical vehicle data to predict future motion statuses within a certain time horizon, aiming to proactively adapt to dynamic traffic conditions. Specifically, the model integrates a seq2seq network with bidirectional long short-term memory (Bi-LSTM) units as the encoder and unidirectional LSTM as the decoder. This adaptation utilizes the robustness and sequential modeling capabilities inherent in LSTM networks. By employing this framework, intricate motion patterns from historical vehicle motion statuses can be effectively extracted and understood, facilitating accurate predictions of future motion statuses. Furthermore, an attention mechanism is introduced to enhance the model's capacity to capture and prioritize critical temporal features, thereby generating context vectors at each time interval. The model is presented in Figure 3.

For a given vehicle $i \in \mathcal{L} \cup \mathcal{F}$, the input sequence over a historical time window of the duration $T$ at a specific time interval $t$ is symbolized as $X_{t-T+1:t}$. For every incremental time step $t'$, each input component is defined as $[v_i(t-t'+1), a_i(t-t'+1)]^T$. The corresponding output from the network is denoted as $Y_{t+1:t+H} = [\hat{a}_i(t+1), \hat{a}_i(t+2), ..., \hat{a}_i(t+H)]$, spanning a prediction horizon $H$.

Then the input sequence is sent to an encoder layer built on the Bi-LSTM structure. The Bi-LSTM mechanism interprets sequential data bidirectionally, leveraging two separate hidden layers conjoined to a unified output. The forward layer iteratively produces the output sequence $\overrightarrow{h}$, using inputs from the time step $t-T-1$ to $t$. Conversely, the backward layer yields the output sequence $\overleftarrow{h}$, with the input sequence inverted, encompassing the interval from $t$ back to $t-T-1$.

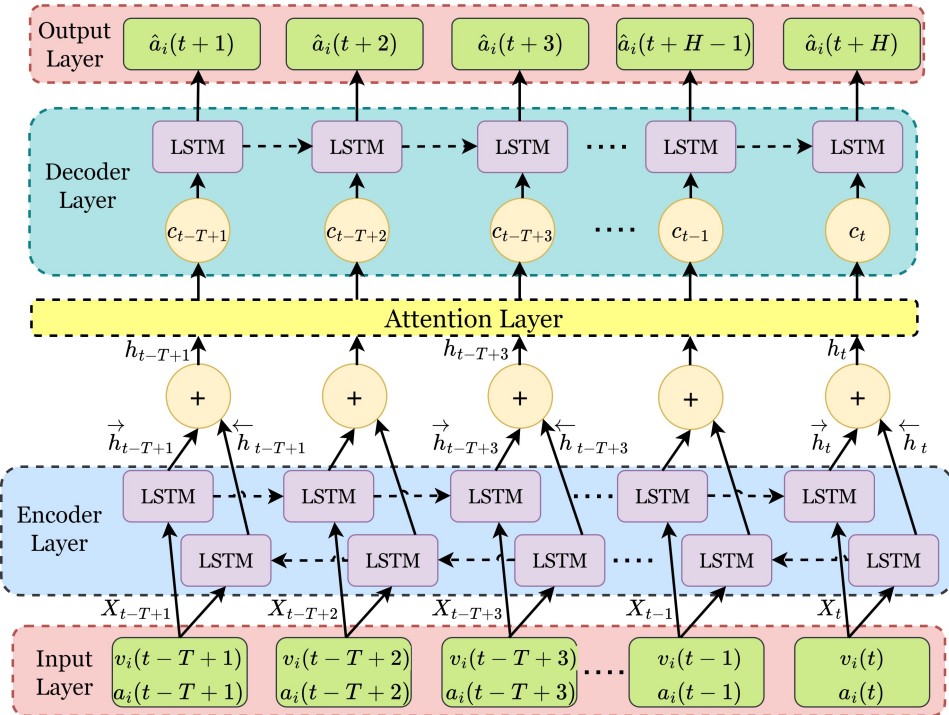

**Figure 3.** Vehicle motion prediction model.

The forward layer outputs are calculated by applying the following updating equations:

$$\overrightarrow{f_t} = \sigma_g(\overrightarrow{W_f}X_t + \overrightarrow{U_f}\overrightarrow{h_{t-1}} + \overrightarrow{b_f}) \tag{1}$$

$$\overrightarrow{i_t} = \sigma_g(\overrightarrow{W_i}X_t + \overrightarrow{U_i}\overrightarrow{h_{t-1}} + \overrightarrow{b_i}) \tag{2}$$

$$\overrightarrow{o_t} = \sigma_g(\overrightarrow{W_o}X_t + \overrightarrow{U_o}\overrightarrow{h_{t-1}} + \overrightarrow{b_o}) \tag{3}$$

$$\overrightarrow{C_t}' = \tanh(\overrightarrow{W_C}X_t + \overrightarrow{U_C}\overrightarrow{h_{t-1}} + \overrightarrow{b_C}) \tag{4}$$

$$\overrightarrow{C_t} = \overrightarrow{f_t} * \overrightarrow{C_{t-1}} + \overrightarrow{i_t} * \overrightarrow{C_t}' \tag{5}$$

$$\overrightarrow{h_t} = \overrightarrow{o_t} * \tanh(\overrightarrow{C_t}) \tag{6}$$

where $X_t$ is the input matrix of the driving behavior; $W_f$, $W_i$, $W_o$, $W_C$, $U_f$, $U_i$, $U_o$, and $U_C$ are the weight matrices; $b_f$, $b_i$, $b_o$, and $b_C$ are the bias vectors; and $\sigma_g$ is the sigmoid gate activation function. The backward layer outputs are calculated in a similar way, and the forward and backward states are concatenated to obtain the sequence ($h_{t-T+1}, h_{t-T+2}, ..., h_{t-1}, h_t$):

$$h_t = [\overrightarrow{h_t}; \overleftarrow{h_t}] \tag{7}$$

Given the heterogeneous distribution of sequence data features, the consequential effect on the output is not uniform. To safeguard the initially input information from attenuation by subsequent inputs during long sequence processing, an attention paradigm is incorporated within the hidden layer. This allows the decoder's input to assimilate diverse intermediate semantics, denoted as *c*. Each *c* is derived through the weight *a* and the output from the Bi-LSTM's encoder hidden layer. By allocating distinct weights to specific values, this paradigm adeptly discerns pivotal features contingent on the input

sequence's influence. Let the state of the decoder's hidden layer be represented by $s_i$; the correlation between $h_j$ and $s_i$ is calculated by applying the additive attention mechanism:

$$e_{ij} = W_1{}^T \tanh(W_2 s_i + U_q h_j) \tag{8}$$

where $W_1$, $W_2$, and $U_q$ are the weight matrices.

The context vector at a certain time step $c_i$ is calculated as follows:

$$c_i = \sum_{j=t-T+1}^{t} \alpha_{ij} h_j \tag{9}$$

$$\alpha_{ij} = \frac{exp(e_{ij})}{\sum_{k=t-T+1}^{t} exp(e_{ik})} \tag{10}$$

where $\alpha_{ij}$ is the weight of the hidden state $h_j$ on the context vector $c_i$.

The decoder iteratively predicts $\hat{a}_i$, and at a decoding time step $j$, the decoder receives the prediction from the previous decoding time step $\hat{a}_i(t + j - 1)$, the context vector $c_j$, and the hidden state $s_{t-1}$. The prediction sequence at each step is computed similarly to the forward layer outputs.

### 3.3. Reinforcement Learning

RL, fundamentally based on the MDP, relies on the continuous interaction between an agent and its environment to autonomously explore optimal behaviors. The core components of RL are defined by the tuple $\mathcal{M} = (S, A, P, R, \gamma)$, where $S$ denotes the state space, $A$ represents the action space, $P : S \times A$ is the state transition function, $R : S \rightarrow \mathbb{R}$ is the reward function, and $\gamma \in (0, 1]$ is the discount factor. The state transition function is given by $p_t(s_{t+1}|s, a) = \mathbb{P}(S_{t+1} = s_{t+1}|S_t = s_t, A_t = a)$. The policy, denoted as $\pi(a|s)$, is a strategy that selects an action based on the observed state from the environment.

In the MDP framework, for a given state $s_t$, the agent tasks an action $a_t$ according to the policy $\pi(a|s_t)$. This action causes the environment to transition to the next state $s_{t+1}$ with the probability $p_t \in P$ and results in the reward $r_t$. Starting from an initial state, the agent repeats this process until reaching a terminal state. The cumulative reward $G_t = \sum_{k=0}^{\infty} \gamma^k R_{t+k}$ is discounted by $\gamma$ to balance immediate and future rewards. The principal objective of RL is to identify the optimal policy $\pi^*$ that maximizes $G_t$. The interaction process between the agent and its environment is depicted in Figure 4.

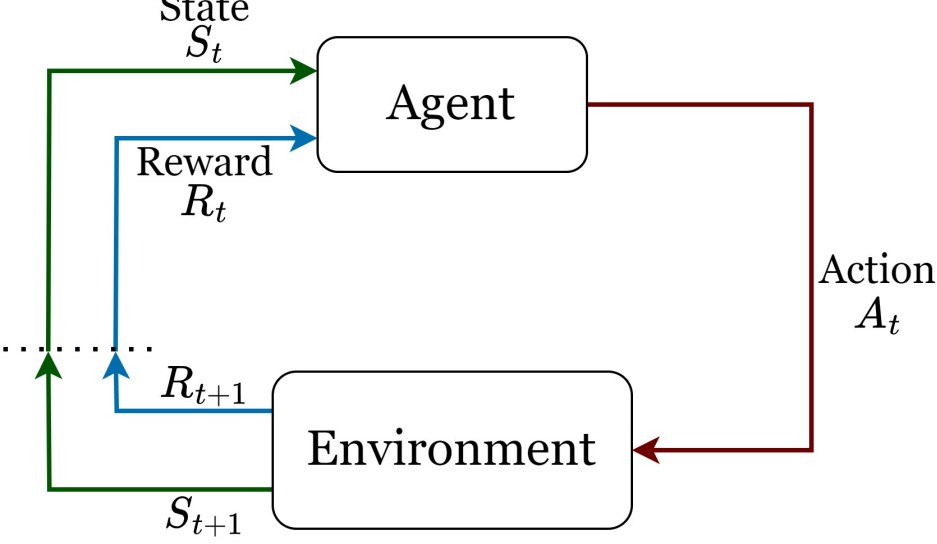

**Figure 4.** Interaction process between agent and environment.

There are mainly three functions in the framework of RL: action-value function, optimal action-value function, and state-value function. The action-value function $Q_\pi$ represents the expected reward given the action $a_t$ taken by the agent when the environment is in the state $s_t$:

$$Q_\pi(s_t, a_t) = \mathbb{E}_{S_{t+1}, A_{t+1}, \dots, S_n, A_n}[G_t | S_t = s_t, A_t = a_t] \tag{11}$$

The state-value function represents the expected reward given the state $s_t$:

$$V_\pi(s_t) = \mathbb{E}_{A_t, S_{t+1}, A_{t+1}, \dots, S_n, A_n}[G_t | S_t = s_t] \tag{12}$$

Both Equations (11) and (12) can be reformulated into the recursive Bellman equation as follows:

$$Q_\pi(s_t, a_t) = \mathbb{E}_{S_{t+1}, A_{t+1}}[R_t + \gamma Q_\pi(S_{t+1}, A_{t+1}) | S_t = s_t, A_t = a_t], \tag{13}$$

$$V_\pi(s_t) = \mathbb{E}_{A_t, S_{t+1}}[R_t + \gamma V_\pi(S_{t+1}) | S_t = s_t] \tag{14}$$

RL algorithms are typically bifurcated into two primary classifications: policy-based and value-based methodologies. The policy is denoted as $\pi_\theta(a|s)$ within policy-based techniques, and the optimal policy is ascertained by fine-tuning parameters $\theta$ to augment the gradient of $\mathbb{E}[G]$. The parameter $\theta$ typically originates from a neural network, known as the policy network, which approximates the policy function $\pi(a|s)$. For a value-based method, the agent achieves the optimal policy by continuously modifying the policy according to updates from Equation (11), with $\pi^* = \arg\max Q_\pi(s_t, a_t)$. It is noteworthy that policy-based techniques, reliant on cyclic updates via the Monte Carlo approach, frequently manifest as algorithmic inefficiencies accompanied by pronounced variance. In contrast, value-based techniques grapple with the intricacies of expansive or continuous action domains. To address these limitations, the actor-critic framework, which merges policy-based and value-based methods, was proposed. In this framework, the actor-network selects actions based on the policy, taking the state as input while outputting the subsequent action. Concurrently, the critic-network evaluates the state-action value function utilizing the action delineated by the actor-network, subsequently furnishing feedback to the actor-network for policy refinement.

### 3.4. Twin-Delayed Deep Deterministic Policy Gradients

The actor-critic paradigm is characterized by its dual-network structure, where an actor network, denoted as $\mu(s; \theta)$, facilitates action determination, and a critic network, expressed as $Q(s, a; \omega)$, undertakes action evaluation. Building on this, the DDPG algorithm emerges as a continuous variant harnessing deterministic policy gradients, resulting in pronounced efficiency by minimizing required training data and bolstering algorithmic convergence [20].

As an evolved variant of DDPG, a TD3 algorithm integrates essential modifications to enhance performance [21]. In order to overcome the overestimation problem of DDPG, the strategy termed clipped double Q-learning is proposed, which incorporates twin value networks $Q(s, a; \omega_1)$, $Q(s, a; \omega_2)$ and one policy network $\mu(s; \theta)$. Each network aligns with a respective target counterpart, namely, $Q(s, a; \bar{\omega}_1)$, $Q(s, a; \bar{\omega}_2)$, and $\mu(s; \bar{\theta})$. The second key enhancement in TD3 is the addition of noise $\xi$, drawn from a clipped normal distribution, to further improve the algorithm's robustness:

$$\bar{a}_{t+1} = \mu(s_{t+1}; \bar{\theta}) + \xi \tag{15}$$

where $\xi \sim \mathcal{CN}(0, \sigma^2, -c, c)$ is a normal distribution with zero mean and standard deviation $\sigma$. The random variable has zero probability of lying outside the interval $[-c, c]$.

Subsequent to this adjustment, dual critic networks collaboratively assess state-action pairs, favoring the lesser computed value to derive the time difference (TD) error $\delta_{i,t}$:

$$\bar{y}_t = \min\{r_t + \gamma Q(s_{t+1}, \bar{a}_{t+1}; \bar{\omega}_i)\} \quad i = 1, 2 \tag{16}$$

$$\delta_{i,t} = Q(s_t, a_t; \omega_i) - \bar{y}_t \quad i = 1, 2 \tag{17}$$

The critic network parameters are updated following the direction of the minimization of the TD error:

$$w_i \leftarrow \underset{w_i}{\operatorname{argmin}} \frac{1}{N} \Sigma_t \delta_{i,t}^2 \quad i = 1, 2 \tag{18}$$

where $N$ is the size of the sample minibatch transition tuples $[s_t, a_t, r_t, s_{t+1}]$ stored in the replay buffer $\mathcal{B}$, which is obtained by implementing the action $a_t$ to the environment at each time step.

The update direction of the actor network $\mu(s; \theta)$ is as follows:

$$\frac{1}{N} \Sigma_t \nabla_a Q(s_t, a; w_1)|_{a=\mu(s_t; \theta)} \nabla_\theta \mu(s_t; \theta) \tag{19}$$

Within the TD3 algorithm framework, the parameters governing the critic networks undergo iterative updates at each computational step. In contrast, the parameters of the actor networks experience periodic updates, specifically at intervals of $K$ steps. This update scheme serves to attenuate the variance inherent in the approximated action value function.

*3.5. Car-Following Control Strategy Construction*

For the CF control strategy, which is constructed under the MDP framework, the state $s_t$ is defined as follows:

$$s_t = [v_0(t), a_i(t), \Delta v_{i,0}(t), \Delta d_{i,0}(t)] \quad \forall i \in \mathcal{L} \cup \mathcal{F} \tag{20}$$

where $v_0(t)$ is the velocity of the SV, $a_i(t)$ is the acceleration of the vehicle $i$, and $\Delta v_{i,0}(t)$ and $\Delta d_{i,0}(t)$ are the relative velocity and distance between the vehicle $i$ and the SV. The exploratory action, provided by the actor network, combines the SV acceleration with added randomness:

$$a_0(t) = \mu(s; \theta) + \epsilon \tag{21}$$

where $\epsilon$ denotes the noise function derived from the Ornstein–Uhlenbeck (OU) stochastic process. As demonstrated by [20], this function enhances the exploratory stochasticity of the action, facilitating the agent's investigation into prospective optimal strategies. The state transition process of $v_0(t)$, $\Delta v_{i,0}(t)$, and $\Delta d_{i,0}(t)$ is calculated as follows:

$$v_0(t+1) = v_0(t) + a_0(t)\Delta t \tag{22}$$

$$\Delta v_{0,h}(t+1) = \Delta v_{0,h}(t) + (v_0(t+1) - v_i(t) - a_i(t)\Delta t) \tag{23}$$

$$\Delta d_{0,h}(t+1) = \Delta d_{0,h}(t) + 0.5(\Delta v_{0,h}(t+1) + \Delta v_{0,h}(t))\Delta t \tag{24}$$

where $v_0(t)$ is the velocity of the SV at the time step $t$.

Another crucial element in constructing the CF strategy is the reward function. It provides the agent with a reward signal, allowing it to discern the quality of its behavior from positive or negative feedback, thereby reinforcing or mitigating its actions. A well-designed reward function enables RL agents to rapidly learn the intended behavior and accelerates convergence during training.

The prediction horizon of the seq2seq model is assumed to be $H$, and for each time step $t \in [t, t + H - 1]$, the reward functions, which account for safety, comfort, energy consumption, and efficiency, are defined as follows:

1. Safety reward function $J_{Safety}$: This study employs the inverse time-to-collision, termed iTTC, as the safety evaluation metric. The risk level is assessed based on a range deemed acceptable to drivers. Accordingly, the safety reward function, founded on this risk threshold, is presented as follows:

$$J_{Safety} = \begin{cases} \log \frac{iTTC_{thr}}{iTTC(t)} & \text{if } iTTC_{thr} > iTTC_0 \\ 0 & \text{otherwise} \end{cases} \tag{25}$$

   where $iTTC_{thr}$ denotes the risk threshold. A more substantial penalty is imposed when $iTTC(t)$ exceeds $iTTC_{thr}$.

2. Comfort reward function $J_{Comfort}$: Jerk is an essential indicator for evaluating ride comfort, which is determined by acceleration variation. Mitigating abrupt changes in jerk during driving minimizes the vehicular inertia felt by passengers, enhancing ride comfort and fuel consumption. The comfort reward function, centered around jerk, is formulated as follows:

$$J_{Comfort} = \beta_2 e^{\beta_1 (\frac{a_0(t) - a_0(t-1)}{\triangle t})^2} \tag{26}$$

3. Energy consumption reward function $J_{Energy}$: To quantify energy usage, this study employs the VT-Micro model, which is rooted in velocity and acceleration parameters, to build the energy consumption reward function:

$$J_{Energy} = \beta_3 e^{\sum_{i=0}^{3} \sum_{j=0}^{3} K_{i,j}(a_0(t)) v_0^i(t) a_0^j(t)} \tag{27}$$

   where coefficients $K_{i,j}(a_0(t))$ depend on the sign of $a_0(t)$ and the values are selected according to [22].

4. Efficiency reward function $J_{Efficiency}$: Fuel consumption tends to increase with the vehicle's velocity. As a result, when the vehicle is moving at higher velocities, it may face significant fuel consumption penalties, making it challenging to maintain a stable time headway. To counterbalance the increased consumption due to higher velocities, a velocity-based reward is introduced to counterbalance the impact of increased velocity, encouraging the vehicle to progress swiftly. The efficiency reward function, developed with this perspective, is presented as follows:

$$J_{Efficiency} = \beta_4 v_0(t) + \beta_5 \log\left(1 + \frac{|T_{0,i}(t) - \beta_6|}{2}\right) \tag{28}$$

   where $T_{0,i}(t)$ is the time headway (thw) based on the ratio of relative distance and SV velocity.

Overall, the combined reward is the aggregate of all of the above-mentioned reward functions:

$$r_t = J_{Safety} + J_{Comfort} + J_{Energy} + J_{Efficiency} \tag{29}$$

To account for future dynamics in the traffic environment, the traditional actor-critic network is augmented as depicted in Figure 5. The actor network provides the acceleration sequence spanning the predictive horizon at the current time step, represented by $a_0(t : t + H - 1)$. The architectures of actor and critic networks consist of an input layer, four fully connected hidden layers, and an output layer. The hidden layers encapsulate neuron counts of 128, 64, 32, and 16 in descending order. The activation functions of these hidden layers are the rectified linear unit (ReLU), whereas the hyperbolic tangent function (tanh) is adopted by the output layer for its activation procedure.

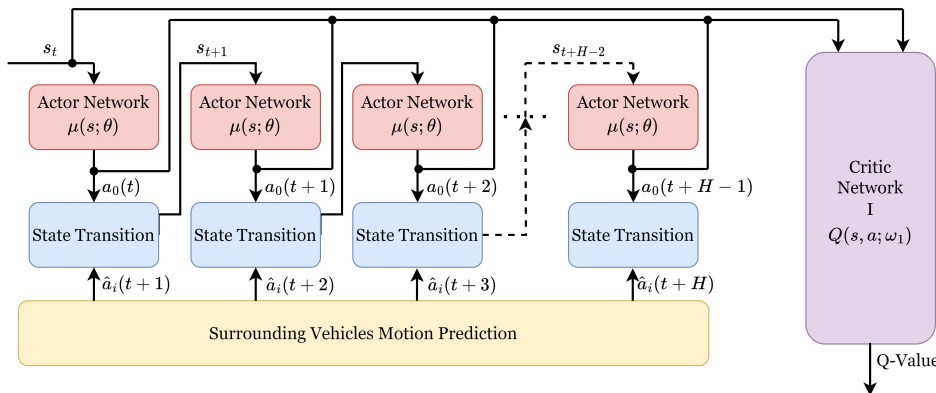

**Figure 5.** Actor-critic network based on surrounding vehicles' motion prediction.

Figure 6 illustrates the training process of the TD3-based actor-critic network. Throughout this process, each time step yields an information tuple $[s_t, a_0(t : t + H - 1), r_t, s_{t+1}]$, which is stored in a replay buffer. Samples from this buffer are subsequently used for updating primary network parameters. In the critic network's update phase, the TD3 algorithm employs target policy smoothing by adding truncated normal distribution noise to target actions. This reduces noise effects on the critic network, bolstering the algorithm's robustness. Furthermore, to mitigate overestimation, the TD3 algorithm uses dual critic networks to independently evaluate state-action pair Q-values and opt for the lesser one. Thus, TD3 encompasses three primary networks: actor, critic I, and critic II networks, along with their corresponding target networks. The primary actor network yields the acceleration sequence $a_0(t : t + H - 1)$, with the reward $r_t$ representing a cumulative value spanning the prediction horizon:

$$r_t = \tilde{r}_t + \sum_{t'=t+1}^{t+H-1} r_{t'} \tag{30}$$

where $\tilde{r}_t$ represents the one-step state transition reward from $s_t$ to $s_{t+1}$ after the action $a_0(t)$. $r_{t'}$ denotes the reward from $s_{t'}$ to $s_{t'+1}$ after implementing $a_0(t')$ within the prediction time horizon based on the surrounding vehicles' motion prediction.

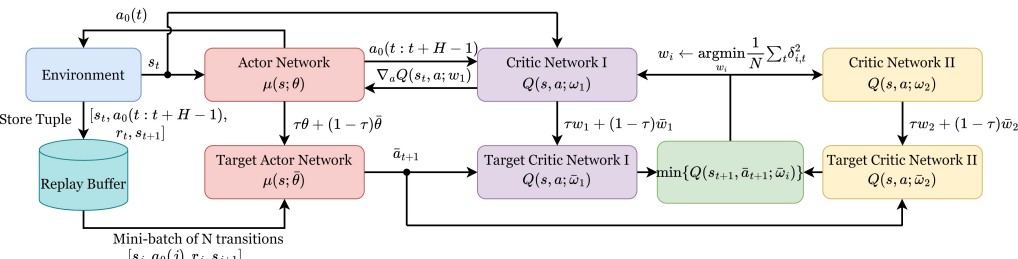

**Figure 6.** Training process of actor-critic network structure based on TD3 algorithm.

The calculation of $r_{t'}$ requires the information $s_{t'+1}$, which is based on the prediction of the future motion statuses of surrounding vehicles. The seq2seq model, utilizing historical kinematic data, is adept at anticipating the accelerations of these surrounding vehicles within the defined predictive horizon. The training process of the TD3 algorithm based on motion prediction is given by Algorithm 1:

---

**Algorithm 1** Training process of motion-prediction-based TD3 algorithm

---

1: Randomly initialize critic networks $Q(s, a; w_1)$, $Q(s, a; w_2)$ and actor network $\mu(s; \theta)$ with random parameters $w_1$, $w_2$ and $\theta$
2: Initialize target networks $\bar{\omega}_1 \leftarrow w_1$, $\bar{\omega}_2 \leftarrow w_2$, $\bar{\theta} \leftarrow \theta$
3: **for** each episode **do**
4:     Obtain the observation state $s_t$ based on the sampled CF scenario
5:     **for** each time step $t$ **do**
6:         Create the action sequence $a_0(t : t + H - 1)$ according to Equation (21) and apply $a_0(t)$ to get $\tilde{r}_t$ after transferring to the new state $s_{t+1}$
7:         Applying the seq2seq motion prediction model to obtain the surrounding vehicles' accelerations within the prediction horizon $\hat{a}_i(t + 1 : t + H)$
8:         **for** $t' = t + 1$ to $t + H - 1$ **do**
9:             Applying $a_0(t')$ to obtain the one-step reward $r_{t'}$ and the transferred new state $s_{t'+1}$
10:         **end for**
11:         Calculating the cumulative reward $\bar{r}_t$ within the time horizon according to Equation (30) and store transition tuples $[s_t, a_0(t : t + H - 1), r_t, s_{t+1}]$ in $\mathcal{RB}$
12:         Sample minibatch of $N$ transitions $[s_j, a_0(j), r_j, s_{j+1}]$ from $\mathcal{RB}$
13:         Target actor network prediction $\bar{a}_0(j + 1) \leftarrow \mu(s_{j+1}; \bar{\theta}) + \xi$
14:         Calculating TD error $\delta_{i,j}$ according to Equations (16) and (17)
15:         Updating critics networks $w_i \leftarrow \underset{w_i}{\arg\min} \frac{1}{N} \sum_j \delta_{i,j}^2$
16:         **if** $t \bmod K == 0$ **then**
17:             Updating the actor network parameter $\theta$ according to the deterministic policy gradient $\frac{1}{N} \sum_j \nabla_a Q(s_j, a; w_1)|_{a=\mu(s_j; \theta)} \nabla_\theta \mu(s_j; \theta)$
18:             Soft updating target actor network $\bar{\theta} \leftarrow \tau\theta + (1 - \tau)\bar{\theta}$
19:             Soft updating target critic network $\bar{w}_i \leftarrow \tau w_i + (1 - \tau)\bar{w}_i$
20:         **end if**
21:     **end for**
22: **end for**

---

## 4. Experimental Results and Analysis

### 4.1. highD Dataset Description

The highD dataset, a cornerstone in the realm of naturalistic driving data, was meticulously compiled to capture an array of vehicular attributes, underpinning the data-driven insights presented in this study. Utilizing advanced aerial surveillance technologies, the dataset encapsulates comprehensive vehicular dynamics, recorded across six distinct segments of a German highway near Cologne over the span of 2017 and 2018. The highD dataset's granular data acquisition, employing drones operating at a 25 Hz recording frequency, has facilitated the assembly of 60 robust records. Each dataset entry meticulously documents a suite of vehicular parameters, including position, lateral and longitudinal velocities, acceleration, and driving direction for an expansive collection of 11,000 vehicles [23].

This rich dataset not only delineates individual vehicle movements with precision but also provides an aggregate representation of traffic flow dynamics, capturing an average duration of 17 min and encompassing an average roadway stretch of 420 m per recording session. The utilization of such precise and expansive data has been pivotal in distilling nuanced understandings of vehicular interactions and driving behaviors, forming the empirical foundation for the CF control strategy developed in this study.

### 4.2. CF Scenario Extraction

Renowned for its comprehensive vehicular data captured via drone technology, the highD dataset underpins the analytical rigor of this research. In this study, the highD dataset is employed to explore vehicular behaviors within a multivehicle context, subtly extending beyond the dataset's traditional utilization (either disregards the trailing vehicle

relative to the SV, as delineated in [17], or just focuses on the immediate leading vehicle, as elucidated in [3]) to encompass a more intricate driving scenario analysis. The investigation centers on a four-vehicle configuration, focusing on interactions involving two leading and one trailing vehicles relative to the SV. This refined analysis framework aims to elucidate the complex vehicle interplay in highway environments, providing a nuanced perspective on vehicular dynamics.

Trajectories of surrounding HDVs were extracted from the highD dataset and kept fixed for the SV during the training process of the RL-based CF strategy. This allows the agent to gain a deeper insight into human driving behavior and identify potential policy enhancements. There are 1612 four-vehicle mode scenarios extracted from the dataset based on the following criteria. The probability distribution histograms of the motion statuses of surrounding vehicles are shown in Figure 7, and kernel density estimation has been employed to approximate the probability density function:

1. The CF behaviors from on-ramp areas are not included.
2. All the vehicles included are cars.
3. The lane changing, cut-in and cut-out behaviors for the vehicles are not included.
4. The duration of CF scenarios is at least 15 s.

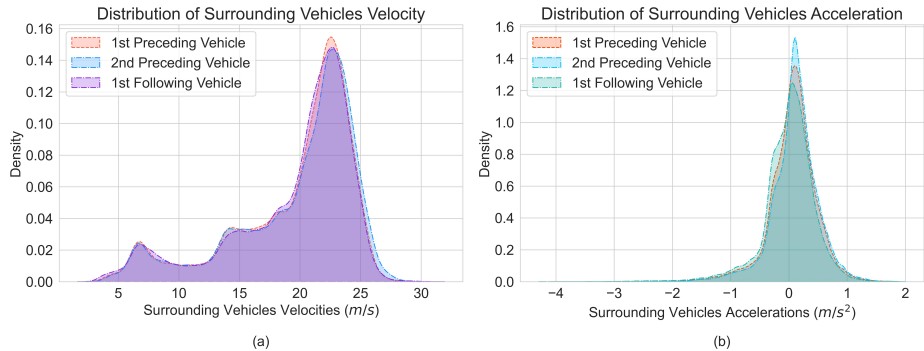

**Figure 7.** Probability density of surrounding vehicles' motion status. (**a**) Velocity. (**b**) Acceleration.

### 4.3. Surrounding Vehicles' Motion Prediction

The parameters for a seq2seq motion prediction network structure are shown in Table 1. The 1612 CF scenarios are utilized for both training and testing the prediction model. These scenarios are divided into 80% for training, 10% for validation, and 10% for testing. The historical feature sequence $I_{t-T+1}$ within the time window is sent to the model at the time step $t$. The model then predicts the sequence $O_{t+1:t+H}$ with the length of $H$, as required by the TD3 algorithm. This process repeats at each subsequent time step. The input sequence shifts by an interval $\triangle t$ until the entire data length for the surrounding vehicles is covered.

**Table 1.** Parameters of seq2seq network structure.

| Parameters | Value |
| --- | --- |
| Encoder hidden units number | 128 |
| Decoder hidden units number | 128 |
| LSTM layer number | 2 |
| Batch size | 256 |
| Learning rate | 0.001 |
| Learning dropout factor | 0.5 |
| Epoch number | 200 |
| Historical input data window length | 40 |
| Prediction target data window length | 20 |

The prediction performance of the seq2seq model on the test dataset for a specific scenario is shown in Figure 8. During the online operation of the RL-based CF strategy, it is imperative for the seq2seq model to precisely predict the kinematic states of surrounding vehicles within the prediction time horizon. As evident from Figure 8, the predicted values for velocity, acceleration, relative distance, and relative velocity are close to the real values. This observation is further corroborated by the mean squared error (MSE) presented in Table 2.

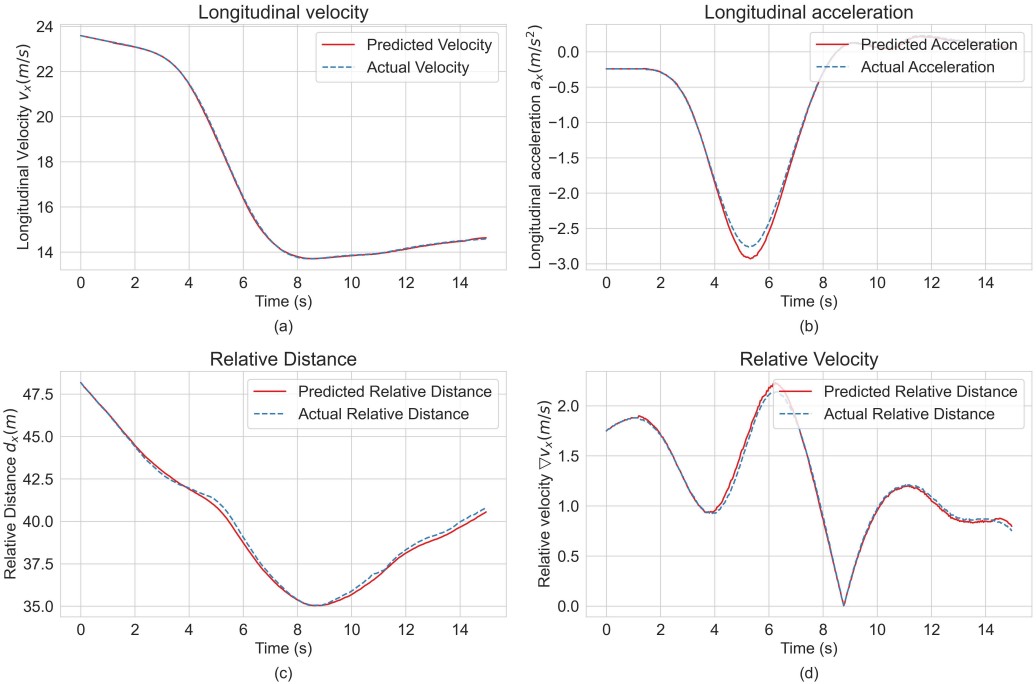

**Figure 8.** The seq2seq prediction performance on the scenario with SV no. 2336 and preceding vehicle no. 2331 of track 26 in the test dataset. (**a**) Longitudinal velocity. (**b**) Longitudinal acceleration. (**c**) Relative distance. (**d**) Relative velocity.

**Table 2.** MSE of prediction result in test dataset.

| Prediction Feature | MSE on Test Dataset |
| --- | --- |
| Velocity (m/s) | 0.3136 |
| Acceleration (m/s$^2$) | 0.0293 |
| Relative distance (m) | 0.6358 |
| Relative velocity (m/s) | 0.3312 |

### 4.4. Validation of TD3 CF Control

For the 1612 extracted CF scenarios, a subset comprising 90% (i.e., 1450) was allocated for training purposes, while the remaining 10% (i.e., 162) was reserved for validation. During training, the agent simulates CF scenarios from the shuffled training data. Once a CF scenario concludes, a new one is randomly selected from the training scenarios, with the agent's state initialized using empirical data from the selected scenario. The training was repeated for 1000 episodes with each episode representing a CF scenario in this context.

The parameters pertinent to the TD3 model can be found in Table 3. In the refinement of the TD3 network, the parameters were meticulously adjusted via the Adam optimizer, capitalizing on the stochastic nature of minibatches to ensure a comprehensive traversal of the solution space. The application of OU-noise, characterized by a mean of 0.15 and a variance of 0.2, was instrumental in augmenting the exploratory capabilities of the action space, thus enhancing the robustness of the model against local optima. This was

complemented by the implementation of Gaussian noise, with a 0 mean and a standard deviation of 0.2, to the target policy, thereby inducing a smoothing effect that further entrenched the stability of the learning process. The learning rates for actor and critic networks were chosen to balance convergence speed and learning stability. A replay buffer of 20,000 and a minibatch size of 256 struck a balance between memory efficiency and sufficient training diversity.

The discount factor at 0.99 ensures a focus on long-term rewards. Soft updates with a coefficient of 0.005 provided gradual target network adjustments. Reward function parameters, such as $\beta_1$ to $\beta_6$, were fine-tuned to accentuate the aspects of safety, comfort, energy efficiency, and overall vehicular efficiency. For instance, the $\beta_1$ and $\beta_2$ parameters in the comfort reward function are configured to penalize high jerk, promoting a driving experience that prioritizes passenger comfort. Similarly, the energy consumption reward function utilizes $\beta_3$ to effectively weigh the energy expenditure against vehicular dynamics. Particularly, the $\beta_4$ and $\beta_5$ coefficients within the efficiency reward function harmonize the intricate relationship between fuel consumption and velocity. The chosen coefficients incentivize fuel-efficient behaviors while also ensuring adherence to optimal time headways, thus reinforcing the synergy between efficiency and safety. Lastly, the threshold $iTTC_{thr}$ was anchored at 0.25, aligned with safety standards and validated through a series of simulations to safeguard against hazardous proximities.

**Table 3.** Parameters of TD3 CF control strategy.

| Parameters | Value |
| --- | --- |
| Learning rate of Adam optimizer for actor networks | $3 \times 10^{-4}$ |
| Learning rate of Adam optimizer for critic networks | $1 \times 10^{-3}$ |
| Replay buffer capacity | 20,000 |
| Minibatch size | 256 |
| Reward discount coefficient | 0.99 |
| Soft update coefficient | 0.005 |
| $iTTC_{thr}$ | 0.25 |
| $\beta_1$ | $-0.06$ |
| $\beta_2$ | 0.5 |
| $\beta_3$ | $-150$ |
| $\beta_4$ | 0.1 |
| $\beta_5$ | $-1.2$ |
| $\beta_6$ | 2 |

Given the variable length of CF episodes and the actor network's random exploration, a moving average of episode rewards with a window size of 50 is utilized to discern reward trends and evaluate the algorithm's performance. This entire process undergoes 10 rounds to evaluate the algorithm's convergence. To verify the performance of the proposed method, some mainstream RL algorithms, such as DDPG, SAC, and PPO, are implemented for comparison. Figure 9 depicts the evolution of the moving average episode reward during the training episodes. Solid colored lines indicate the mean over multiple rounds, while shaded regions denote the standard deviation around these mean values. The TD3 agent's moving average episode reward outperforms those of other agents, including those involving human driving actions. The TD3 algorithm's reward converges around 820, whereas the SAC reward stabilizes around 635. Both the PPO and DDPG algorithms consistently underperform compared with human driving.

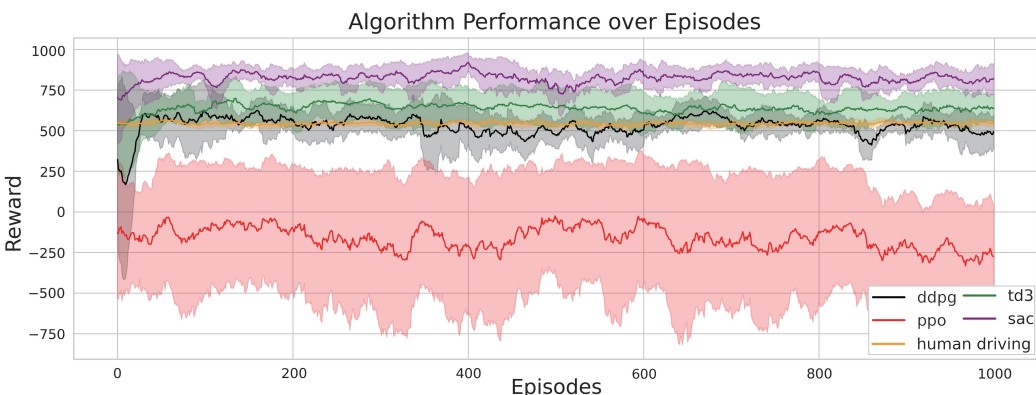

**Figure 9.** Algorithm performance over episodes.

TD3's dual Q-learning mechanism aims to mitigate overestimations of Q-values, while its delayed policy updates ensure infrequent policy modifications. Furthermore, TD3 introduces a policy noise clipping technique, preventing excessive variations during policy updates. In continuous action scenarios like CF, vehicles are expected to respond smoothly and consistently across varying driving scenarios. Any abrupt or substantial policy shifts can lead to suboptimal driving behavior. In comparison, DDPG may suffer from overestimation of Q-values, leading to unstable training. While PPO exhibits commendable performance in discrete action space tasks, it may not be as adept as TD3 or SAC in continuous control tasks. On the other hand, while SAC enhances its policy's exploratory nature using the maximum entropy principle and is generally effective, its stochastic approach can sometimes lead to undue exploration. This might cause SAC to underperform relative to TD3 in certain driving scenarios. Subsequent analysis will thus center on comparing the TD3 agent's performance with human driving behavior.

To juxtapose the performance of the TD3 policy with human driving behavior, the quantitative evaluation results are presented in Table 4. The safety check index represents the ratio of the CF scenarios where $iTTC$ surpasses a given threshold to the total number of CF scenarios. The energy consumption index of the SV is calculated based on Equation (27). The other indices are the average results calculated based on the whole test dataset.

**Table 4.** Quantitative evaluation results on test dataset.

| Evaluation Index | TD3 | highD |
|---|---|---|
| Average velocity of SV (m/s) | 22.5929 | 20.2627 |
| Relative distance with 1st preceding vehicle (m) | 25.1463 | 29.5840 |
| Relative Distance with 1st following vehicle (m) | 35.8522 | 29.8769 |
| Relative Velocity with 1st preceding vehicle (m/s) | 0.1741 | 1.1348 |
| Relative velocity with 1st following vehicle (m/s) | 0.0432 | 0.0206 |
| Safety check | 0.63% | 6.47% |
| Time headway (s) | 1.7021 | 2.4786 |
| Energy consumption index of SV | −4.8632 | −6.3158 |
| Jerk (m/s$^3$) | 0.0532 | 0.2107 |

As indicated in Table 4, the penalty imposed during TD3 agent training for exceeding the risk threshold $iTTC_{thr}$ results in only 0.63% of scenarios surpassing $iTTC_{thr}$, which is lower than that of the highD dataset. Additionally, Table 4 highlights that the TD3 agent's average velocity surpasses that of the highD dataset. With a focus on safety, the TD3-controlled agent not only maintains a reduced relative distance to the preceding vehicle but also ensures sufficient acceleration room for the following vehicle. This observation underscores the significant efficacy of the TD3-controlled agent in enhancing the efficiency of CF behaviors. Figure 10 compares the performances of the TD3 agent and human driving

in a specific scenario. It reveals that the TD3-controlled SV excels in maintaining a steady traffic flow, especially in the relative velocity between the preceding and following vehicles. This is due to the seq2seq model's prediction for the motion status of the surrounding vehicles within the prediction horizon. Such anticipatory capabilities enable the model to incorporate the prospective motion trajectory of the SV during acceleration determinations at each iterative step, thereby circumventing potential compromises in CF safety and efficiency due to abrupt accelerations or decelerations exhibited by the preceding vehicle. On the other hand, the jerk exhibited by the TD3-controlled agent is considerably diminished relative to human drivers. This not only enhances ride comfort but also results in a better time headway compared with human drivers.

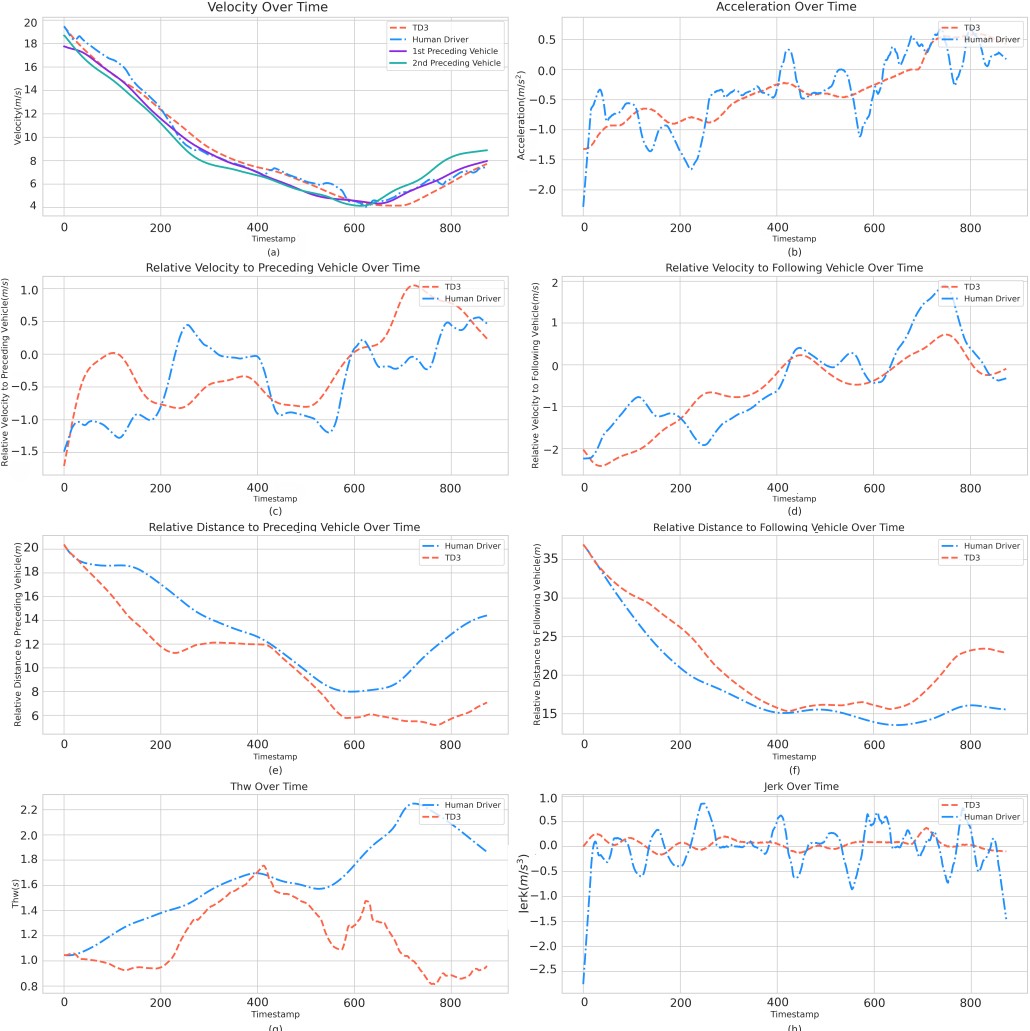

**Figure 10.** Comparison of performances between the TD3 agent and human driving on the scenario with SV no. 2742, 1st preceding vehicle no. 2738, 2nd preceding vehicle no. 2736, and 1st following vehicle no. 2744 of track 25 in the test dataset. (**a**) Velocity. (**b**) Acceleration. (**c**) Relative velocity to preceding vehicle. (**d**) Relative velocity to following vehicle. (**e**) Relative distance to preceding vehicle. (**f**) Relative distance to following vehicle. (**g**) Thw. (**h**) Jerk.

### 4.5. Platoon Analysis

Given the stable highway conditions in the highD dataset, in order to fully exploit the advantages of TD3-controlled agents in traffic oscillation multivehicle scenarios, the platoon analysis, which consists of nine vehicles with one head vehicle, is performed. The optimal velocity model (OVM) is utilized to depict the dynamics of HDVs, and the parameters are referred to [24]. The first simulation examines a traffic wave scenario,

induced by introducing a sinusoidal velocity perturbation to the head vehicle around an equilibrium velocity of 15 m/s. Within the platoon, two CAVs are positioned as the fourth and seventh vehicles. The model predictive control (MPC) controllers for CAVs are implemented according to [24] to compare the performances of the TD3-controlled CAVs. The control objective for the platoon is to stabilize the velocity of each vehicle around 15 m/s, while maintaining the intervehicle spacing close to 20 m. The acceleration constraints for the CAVs range from $-5$ m/s$^2$ to 2 m/s$^2$, while the intervehicle spacing constraints range from 5 m to 40 m . The results are shown in Figure 11, where the first row represents the scenario where all vehicles in the platoon are HDVs and the second and third rows, respectively, illustrate the performance of the platoon that includes two CAVs controlled by the MPC controller and the TD3 algorithm.

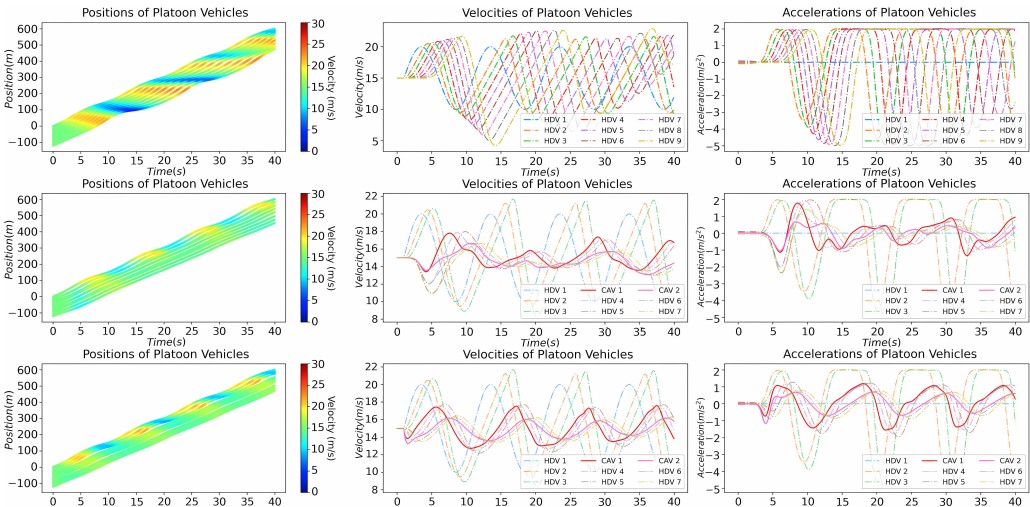

**Figure 11.** Sinusoidal perturbation platoon analysis. The first row in the figure represents the all-HDV scenario, while the second and third rows represent the MPC-controlled CAVs and TD3-controlled CAVs, respectively.

Figure 11 illustrates that as the sinusoidal perturbation wave propagates backward through the platoon, its amplitude amplifies, leading to a significant increase in overall fuel consumption and collision risk. In contrast, when the platoon incorporates CAVs controlled by either the MPC or TD3 algorithm, the amplitude of the traffic shockwave is effectively attenuated, which underscores their capabilities in mitigating disturbances and stabilizing the traffic flow. Furthermore, the quantitative evaluation results between the MPC controller and the TD3 algorithm are shown in Table 5. It is worth noting that MPC employs precise linearized dynamics around the equilibrium state for control input design, and it can be served as a valuable benchmark for comparison, while the TD3 algorithm directly relies on the raw data obtained from the environment to perform the self-exploration strategy. As Table 5 indicates, TD3-controlled CAVs slightly outperform MPC in noise-perturbed nonlinear traffic systems, even in the absence of explicit system knowledge. Specifically, although CAVs controlled by the TD3 algorithm slightly decreased the platoon's overall average speed in comparison with those controlled by MPC, the relative velocities and accelerations are reduced. In the meantime, the relative distance between vehicles closer to the equilibrium state and the enhanced ride comfort are sustained. This highlights the capability of TD3-controlled CAVs to achieve a more stable and fuel-efficient traffic flow.

**Table 5.** Evaluation results on sinusoidal perturbation scenario.

| Evaluation Index | TD3 | MPC |
|---|---|---|
| Average velocity (m/s) | 14.8051 | 14.9559 |
| Average relative velocity (m/s) | 0.4960 | 0.5314 |
| Average accelerations (m/s$^2$) | 0.4640 | 0.5837 |
| Average relative distance (m) | 18.4775 | 15.8521 |
| Average jerk (m/s$^3$) | 0.0063 | 0.0103 |

In the second scenario, an emergency braking event is simulated to assess the safety efficacy of TD3-controlled CAVs. Initially, the head vehicle in the platoon sustains an equilibrium velocity of 15 m/s for the first second. It then abruptly decelerates at a rate of $-5$ m/s$^2$ for the subsequent 2 s and reaches a lower velocity of 5 m/s. Subsequently, it maintains this lower velocity for 6 s. Afterward, it accelerates back to its original velocity at a rate of 2 m/s$^2$ and maintains that velocity for the remaining time. The results are shown in Figure 12.

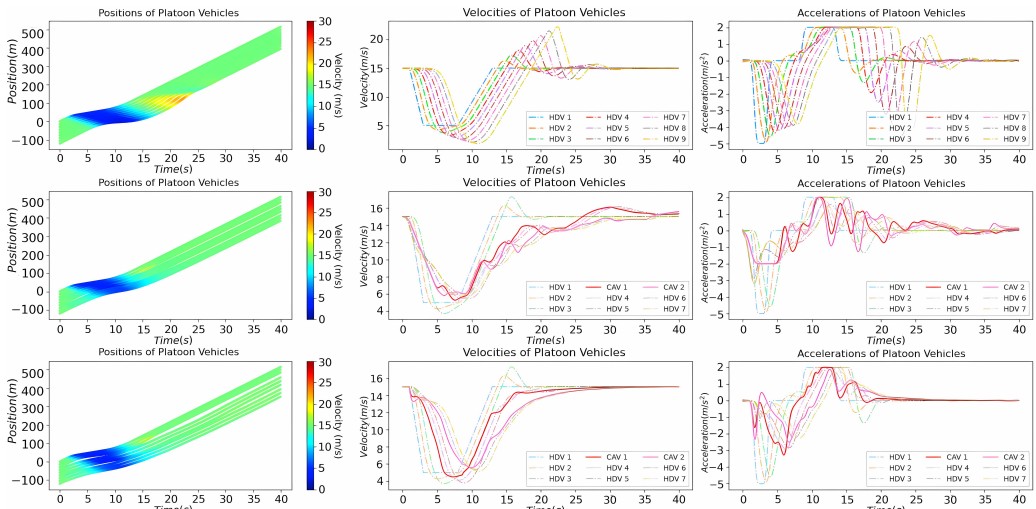

**Figure 12.** Braking perturbation platoon analysis. The first row in the figure represents the all-HDV scenario, while the second and third rows represent the MPC-controlled CAVs and TD3-controlled CAVs, respectively.

In Figure 12, it is evident that with an all-HDV platoon, significant velocity fluctuations arise due to the braking perturbation of the lead vehicle. However, introducing two CAVs controlled by MPC and the TD3 algorithm results in a significant attenuation of the undesired traffic wave. Upon detecting the lead vehicle's braking, these CAVs promptly decelerate, thereby ensuring a safer following distance. When comparing the performances of the MPC controller and the TD3 algorithm, the TD3 algorithm yields more stable velocity and acceleration profiles, while MPC demonstrates some overshoots. The quantitative assessments presented in Table 6 further substantiate these observations.

**Table 6.** Evaluation results on braking perturbation scenario.

| Evaluation Index | TD3 | MPC |
|---|---|---|
| Average velocity (m/s) | 12.2833 | 12.7025 |
| Average relative velocity (m/s) | 0.3838 | 0.4323 |
| Average accelerations (m/s$^2$) | 0.5255 | 0.6101 |
| Average relative distance (m) | 17.6641 | 14.2603 |
| Average jerk (m/s$^3$) | 0.0016 | 0.0142 |

## 5. Conclusions

In this study, a CF control strategy leveraging the reinforcement learning TD3 algorithm is proposed. The CF scenario considered is the four-vehicle mode, which includes two vehicles ahead of the SV and one vehicle behind. To encapsulate the dynamic uncertainties inherent in CF control, a seq2seq predictive module is integrated. Modifications to the TD3 algorithm ensure that the strategy is adaptive to potential uncertainties in future traffic conditions, which improves the robustness of the algorithm. The algorithm is first validated in the environment composed of the trajectories extracted from the naturalistic driving dataset to simulate the mixed traffic flow of HDVs and CAVs. The reward function encompassing the safety, comfort, efficiency, and fuel consumption metrics is designed for guiding the agent navigate through the environment. The training and testing results of the improved TD3 algorithm are compared with some typical reinforcement learning algorithms based on the trajectories sampled from the highD dataset, and the results indicate superior reward convergence with the TD3 algorithm. Subsequently, two traffic perturbation scenarios are implemented to further demonstrate the capabilities of a mixed platoon with TD3-controlled CAVs. The results show that CAVs effectively mitigate undesired traffic waves stemming from head vehicle perturbations while maintaining the desired traffic states. Finally, when compared with the performance of the MPC-controlled CAVs, the TD3-controlled CAVs consistently sustain a more stable and efficient traffic flow.

Future improvements to this work could include the following aspects: First, the reward function can be extended to represent different types of human drivers, such as aggressive or conservative drivers. Their distinct perceptions of the environment and driving objectives necessitate specific strategies for agents interacting with them. This study primarily focuses on scenarios with relatively stable motion statuses of the surrounding vehicles, but in an actual traffic environment, more intricate traffic situations like the lead vehicle changing lanes, the cut-in behaviors of the vehicles in the adjacent lanes, and overtaking maneuvers should be considered. Lastly, the highD dataset primarily provides stable CF scenarios, and an aggregation of popular naturalistic driving datasets currently available could facilitate agent training across a wider range of scenarios and enhance the performance of the agent in diverse situations.

**Author Contributions:** Conceptualization, T.W. and D.Q.; methodology, T.W.; software, T.W. and K.W.; validation, S.D.; formal analysis, T.W. and K.W.; investigation, S.D.; resources, K.W.; data curation, S.D.; writing—original draft preparation, T.W.; writing—review and editing, T.W.; visualization, K.W. and S.D.; supervision, D.Q.; project administration, T.W. and K.W.; funding acquisition, D.Q. All authors have read and agreed to the published version of the manuscript.

**Funding:** The project was supported by the National Natural Science Foundation of China (Grant No. 52272311).

**Institutional Review Board Statement:** Not applicable.

**Informed Consent Statement:** Not applicable.

**Data Availability Statement:** Data are contained within the article.

**Conflicts of Interest:** The authors declare no conflicts of interest.

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
