# Peer review of "Deep Reinforcement Learning Car-Following Control Based on Multivehicle Motion Prediction"

_electronics, doi:10.3390/electronics13061133_

Round 1

Reviewer 1 Report

Comments and Suggestions for Authors

The paper discusses the intresting topic and provides machine learning-based solution to predict the status of surrounding vehicles for autonomous vehicles.
The discussed the problem and the proposed approach rigorously.
I recomend the auhtors to include the follwoing in the paper

- Please provide the reference or provide short description highD dataset/
- It is better to discuss the compexity of the algorithm.

- Please provide the justification of parameter values in Table 3.

Comments on the Quality of English Language

Read manuscript carefully for minor editing. 

Reviewer 2 Report

Comments and Suggestions for Authors

Increasing safety is a major task for most of the road traffic administrators. Any study about revolutionary ways to reach this target is welcome. To have a good and clear policy for safety means to understand all participants behaviours. 

During times different approaches have developed to get a more clear image about road traffic characteristics. In this way, academia plays an important role through their researches.

The present paper describe one of the methods that can be use for this purpose. Paper structure is well defined and allow to the reader a clear understanding of the study. Experiments are clear and analyse of the results validate the values. 

Conclusions are based on the study results and include the future research intended to be developed by the authors, possible to improve the present paper.

Paper quality is good, equations are clear and explained.

Some of the figures are recommended to be improved, figures 9, 10, 11 and 12.

Reviewer 3 Report

Comments and Suggestions for Authors

This article presents a reinforcement learning based car following strategy. The analysis was performed on the highD dataset. Overall, the article presented is interesting, yet I have the following comments:.

1. Please revise the article, i noticed a typo in figure 2: "reward."

2.  You need to have a subsection in the experimental results giving more details about the structure of the highD dataset. The information mentioned, starting at line 350, is not detailed. What is the normal use of this dataset, Could other research proposed on it be compared to yours, or are you employing it in a different manner than the other research?

3. You need to update some of the statements in the article to show what your actual work is. For example, figure 2 Is this "the proposed car-following control framework" or is it just a generic, well-known one? Given that it does not include a reference, i am assuming it is yours, yet I am confused why it is not in the methodology section.

4. Concerning the vehicle motion prediction framework. Again, is this your proposed model? The seq2seq is a well known model. So, if you are using it as it is where is the reference and if you updated it then what are the updates? 

5. I am mostly confused because it is hard to tell what is actually the contribution and what is merely a background. The focus should be mostly on the contribution rather than just repeating history

6. Finally, are these results comparable to others? 

7. The limitation and contribution parts are too long, making it hard to focus on the actual work.

Round 2

Reviewer 3 Report

Comments and Suggestions for Authors

The paper is much improved after the modifications. Well done.